# First Detection of Influenza D Virus Infection in Cattle and Pigs in the Republic of Korea

**DOI:** 10.3390/microorganisms11071751

**Published:** 2023-07-05

**Authors:** Eui Hyeon Lim, Seong-In Lim, Min Ji Kim, MiJung Kwon, Min-Ji Kim, Kwan-Bok Lee, SeEun Choe, Dong-Jun An, Bang-Hun Hyun, Jee-Yong Park, You-Chan Bae, Hye-Young Jeoung, Kyung-Ki Lee, Yoon-Hee Lee

**Affiliations:** 1Viral Disease Division, Animal and Plant Quarantine Agency, 177 Hyeoksin 8-ro, Gimcheon 39660, Republic of Korea; sksms147zld@naver.com (E.H.L.); saint78@korea.kr (S.-I.L.); mingming0525@korea.kr (M.J.K.); alwjd3920@naver.com (M.K.); mj852@hanmail.net (M.-J.K.); ivvi59@korea.kr (S.C.); andj67@korea.kr (D.-J.A.); hyunbh@korea.kr (B.-H.H.); 2Chungnam Veterinary Research Institute, 37 Gulpo-gil, Taean 32138, Republic of Korea; kbs97330@naver.com; 3Import Risk Assessment Division, Animal and Plant Quarantine Agency, 177 Hyeoksin 8-ro, Gimcheon 39660, Republic of Korea; parkjyyunesy@korea.kr; 4Animal Disease Diagnostic Division, Animal and Plant Quarantine Agency, 177 Hyeoksin 8-ro, Gimcheon 39660, Republic of Korea; kyusfather@korea.kr (Y.-C.B.); jhy98@korea.kr (H.-Y.J.); naturelkk@korea.kr (K.-K.L.)

**Keywords:** influenza D virus, HEF gene, D/Yamagata/2019 lineage

## Abstract

Influenza D virus (IDV) belongs to the Orthomyxoviridae family, which also include the influenza A, B and C virus genera. IDV was first detected and isolated in 2011 in the United States from pigs with respiratory illness. IDV circulates in mammals, including pigs, cattle, camelids, horses and small ruminants. Despite the broad host range, cattle are thought to be the natural reservoir of IDV. This virus plays a role as a causative agent of the bovine respiratory disease complex (BRDC). IDV has been identified in North America, Europe, Asia and Africa. However, there has been no information on the presence of IDV in the Republic of Korea (ROK). In this study, we investigated the presence of viral RNA and seroprevalence to IDV among cattle and pigs in the ROK in 2022. Viral RNA was surveyed by the collection and testing of 999 cattle and 2391 pig nasal swabs and lung tissues using a real-time RT-PCR assay. IDV seroprevalence was investigated by testing 742 cattle and 1627 pig sera using a hemagglutination inhibition (HI) assay. The viral RNA positive rate was 1.4% in cattle, but no viral RNA was detected in pigs. Phylogenetic analysis of the hemagglutinin-esterase-fusion (HEF) gene was further conducted for a selection of samples. All sequences belonged to the D/Yamagata/2019 lineage. The seropositivity rates were 54.7% in cattle and 1.4% in pigs. The geometric mean of the antibody titer (GMT) was 68.3 in cattle and 48.5 in pigs. This is the first report on the detection of viral RNA and antibodies to IDV in the ROK.

## 1. Introduction

Influenza viruses are segmented, enveloped RNA viruses that belong to the family *Orthomyxoviridae* and are divided into four types (A, B, C and D) [1]. Influenza A, B and C viruses have been documented to infect mammals, humans and other species [2]. Influenza A virus (IAV) in particular infects birds, pigs and humans, and there have been reports of transmission from wild birds to pigs and from pigs to humans. It has been recognized as having contributed to the global influenza pandemic [3]. Influenza B virus (IBV) also infects humans [4]. Both IAV and IBV have been reported to cause infectious diseases with high lethality in humans and cause influenza epidemics. Structurally, IAV and IBV both contain eight genomic segments including two surface glycoproteins, neuraminidase (NA) and hemagglutinin (HA) [5]. On the other hand, the genome of influenza D virus (IDV), like influenza C virus (ICV), incorporates seven RNA segments. Even so, IDV shares less than 50% protein sequence identity with ICV, the most genetically similar member of the influenza virus family [6,7]. IDV was first isolated from pigs in Oklahoma, United States, in 2011 [7]. Since then, IDV has been detected worldwide, in various ranges of hosts. After its first isolation, IDV was found in countries in North America (United States, Canada), Europe (France, Ireland, Italy, the United Kingdom and Luxembourg), Asia (China and Japan) and Africa (Ethiopia, Togo and Kenya) [8,9,10].

In Asia, IDV was first detected in 2014 in Shandong, China, in samples from healthy cattle [11]. This was followed by the first case of IDV in Japan in 2016, which was identified from a group of cattle in the Ibaraki Prefecture [12]. Recently, in 2020, a case of IDV infection in cattle was reported for the first time in Turkey [13]. The continued reports of IDV detection in various countries supports the view of worldwide spread of IDV. Despite the growing evidence, there have been no reports of IDV presence in the ROK.

IDV can infect and spread to a wide range of hosts, including domestic cattle, pig, sheep, goats, horses, camels and poultry, as well as wild species such as feral pigs [9,10,14]. Cattle are considered to be the main reservoir of IDV, with infection being associated with mild to serve respiratory clinical signs (e.g., cough, discharge from the nose and dyspnea) [15]. The recently discovered IDV is thought to induce respiratory illness primarily in cattle and to a much lesser extent in pigs [16]. The zoonotic potential has not yet been fully elucidated, but the serological and virological tests have revealed that it could infect humans. IDV can be genetically classified into six lineages so far: D/OK (D/swine/Oklahoma/1334/2011), D/660 (D/bovine/Oklahoma/660/2013), D/Yama2016 (D/bovine/Yamagata/10710/2016), D/Yama2019 (D/bovine/Yamagata/1/2019), D/CA2019 (D/bovine/California/0363/2019) and D/Bursa2013 (D/bovine/Turkey-Bursa/ET-130/2013), based on the hemagglutinin-esterase-fusion (HEF) gene, which is the major target for the neutralizing antibodies generated during IDV infection [17].

In 2022, Korean native cattle and non-indigenous beef cattle represented 90% (3,693,779 head) of the total cattle population (4,083,767 head), while 10% (389,988 head) were dairy cattle. The small proportion of dairy cattle in the national herd can be attributed to dairy products being of less importance in the traditional Korean diet. Based on the population size of the farms, the percentage of cattle farms (total of 97,413 farms) that were raising fewer than 20 head, 20 to 50 head, 50 to 100 head and more than 100 head were 48% (46,521 farms), 26% (24,947 farms), 16% (15,557 farms) and 11% (10,388 farms), respectively. In 2022, more than ten million pigs (11,123,872 head) were raised in the Republic of Korea. Based on the population size of the farms, the percentage of pig farms (total of 5696 farms) rearing fewer than 1000 head, 1000 to 5000 head, 5000 to 10,000 head, and more than 10,000 head were 41% (2345 farms), 52% (2942 farms), 5% (306 farms) and 2% (104 farms), respectively [18]. According to the study report by the Ministry of Agriculture and Forestry conducted from 30 June through 3 November 2003, it was found that 43,383 farms, which is 22% of the total number of cattle farms (201,661) during the period, were raising cattle with other livestock. However, only 1% (606 farms) were rearing cattle and pigs together, showing that it was a rare practice [19].

Based on the growing evidence of IDV infection around the world, including in the neighboring countries of China and Japan, we conducted a study to investigate the possible presence of IDV in the ROK by testing for viral RNA and antibodies of IDV in cattle and pigs. Samples were collected from the provinces of Gyeonggi, Gangwon, Chungbuk, Chungnam, Jeonbuk, Jeonnam, Gyeongbuk, Gyeongnam and Jeju Island, which is the largest island located in the southernmost part of Korea in 2022.

## 2. Materials and Methods

### 2.1. Sampling Strategy

From January to December 2022, totals of 742 cattle sera (644 farms), 954 cattle nasal swabs (801 farms), 1627 pig sera (88 farms) and 2232 pig nasal swabs (287 farms) were collected from abattoirs in the ROK. We also collected 45 cattle lung tissues (37 farms) and 159 pig lung tissues (68 farms) submitted for diagnostic purposes to the Animal and Plant Quarantine Agency (APQA) in 2022. This included pig lung samples showing gross pathology indicative of streptococcal pneumonia infection and respiratory disease. In addition, one hundred feral pig sera collected for survey of classical swine fever (CSF) were received from the CSF laboratory, Viral Disease Division, of the APQA. Nasal swab and blood samples were collected using NS—1 nasal swab (Noble Biosciences, Inc., Gyeonggi-do, Republic of Korea) and BD vacutainer K2E (EDTA) (BD Biosciences, Franklin Lakes, NJ, USA), respectively. Samples were stored at −80 °C until further analysis. 

### 2.2. Reference Viruses

Reference viruses from the prototypal D/OK lineage, D/Swine/Oklahoma/1334/2011 [7] and D/bovine/France/5920/2014 [20] were kindly provided by the Animal Plant Health Agency (APHA), United Kingdom. The viruses were propagated in Madin-Darby canine kidney (MDCK) cells as previously described [21].

### 2.3. Hemagglutination Inhibition Assay

Pig serum samples were pretreated overnight at 37 °C with receptor-destroying enzyme (RDE; Denka, Tokyo, Japan) at a ratio of 1:3. Cattle serum samples were not RDE treated. Hemagglutination assay (HA) and hemagglutination inhibition (HI) assay were performed as previously described for bovine serum [22] and in accordance with the WOAH manual for swine influenza A virus infection for pig serum [23]. All serum samples were heat treated for 1 h at 56 °C in a water bath [22]. Twofold diluted serum (25 μL) and the same volume of 4 hemagglutination (HA) units of reference virus (IDV) were mixed in a 96-well microplate with a V-shaped bottom and incubated at room temperature for 30 min. Then, 25 μL of 1% chicken red blood cells (RBCs) were added to each well and incubated at 4 °C for 1 h. The HI titer was determined as the reciprocal of the end-point dilution that showed complete HI. The HI titer ≥ 20 was used as the cut-off value for seropositive in all samples [22,24,25]. We tested pig serum samples using the reference virus D/Swine/Oklahoma/1334/2011, and cattle serum samples using D/bovine/France/5920/2014.

### 2.4. Real-Time RT-PCR

Nasal swab and lung tissues were prepared using the RNeasy mini kit (Qiagen, Hilden, Germany) following the manufacturer’s instructions. Viral nucleic acids were extracted from a separate aliquot of each nasal swab and lung tissue. Nucleic acids were tested for PB1 gene of IDV using a Step One Plus Real-Time PCR System (Applied Biosystems, Foster City, CA, USA) using previously reported primer and probe; their sequences were as follows: forward primer: 5′-GCT GTT TGC AAG TTG ATG GG-3′; reverse primer: 5′-TGA AAG CAG GTA ACT CCA AGG-3′; and probe: 5′ FAM-TTC AGG CAA GCA CCC GTA GGA TT-BHQ1 3′ [7].

### 2.5. Genomic Sequencing and Phylogenetic Analysis

cDNA was prepared from each RNA sample using SuperScript Ⅲ First-Strand Synthesis System for RT-PCR (Invitrogen, Carlsbad, CA, USA). Then, cDNA was amplified at 50 µL volume using Accupower ProFi Taq PCR Premix (Bioneer, Oakland, CA, USA) with universal primer [26,27]. The HEF full gene was amplified using the One-Step reverse transcription (RT)-PCR kit (Qiagen, Germantown, MD, USA) and was sequenced by a commercial Sanger sequencing facility (Macrogen, Seoul, Republic of Korea). Reference sequences of the HEF full gene were downloaded from the GenBank database (NCBI: http://ncbi.nlm.nih.gov (accessed on 20 December 2022) for phylogenetic analysis. The sequences were aligned and assembly was performed with ClustalW in MEGA 11 [11].

### 2.6. Data Analysis

Data were recorded in a standard Excel spreadsheet in Microsoft Excel 2016 (Microsoft corporation, Seattle, WA, USA). The geometric mean antibody titer (GMT) was calculated to compare the level of HI titer in positive sera only. GMT was calculated by averaging the logarithms of the HI titer and then converting the mean to a real number. Positive rates of animals by region and species are the number of positive animals divided by the total number of tested animals. The apparent herd prevalence was calculated as the total number of positive farms divided by the total number of tested farms.

## 3. Results

### 3.1. Seropositivity of IDV in Cattle and Pigs

Of the 742 serum samples collected from 644 cattle farms, 405 serum samples (54.6%) from 364 farms (56.5%) were positive by the HI assay. As shown in Table 1, the positive rate (%) was the highest in Gyeonggi Province at 84%, followed by Chungbuk, Chungnam, and Jeonbuk Provinces at 75%, 73.6% and 73.5%, respectively. No positive sample was found in Jeju Island. We confirmed that the antibodies to IDV were widespread in cattle in the Korean peninsula except Jeju Island. The GMT were in the range of 50.5~113.8 (mean 68.3) in seropositive samples. The levels of GMT was the highest in Jeonnam Province (113.8). The HI titers observed in positive samples were in the range of 32~2048. Distribution of HI titer 32~64 was 70.1% (Figure 1). Of the 1627 serum samples collected from domestic pigs on 88 farms, a total of 23 serum samples (1.4%) in 13 farms (14.8%) were positive in the HI assay. As shown in Table 2, the positive rate (%) was the highest in Chungbuk Province at 6.3%. No positive samples were found in Gyeonggi, Gyeongnam, Jeonbuk or Jeonnam Provinces or Jeju Island. In domestic pigs, antibodies to IDV were detected in Gangwon, Chungbuk, Chungnam and Gyeongbuk Provinces. The GMT were in the range of 32~55.7 (mean 50.3) in seropositive samples. The HI titers observed in the positive samples were in the range of 32~512. Distribution of HI titer 32~64 was 95.6% (Figure 2). We also investigated antibodies of IDV in feral pigs. There were no positive samples out of the 100 sera tested. As a result, it was confirmed that domestic pigs were seropositive for IDV, but IDV antibodies were not detected in the small number of feral pig samples analyzed.

### 3.2. Molecular Detection of IDV RNA in Cattle and Pig Samples

A total of 999 nasal swabs and lung tissues (838 farms) from cattle were tested by real-time RT-PCR assay. IDV RNA was detected in 14 nasal swabs (1.4%) from samples and specifically in 14 nasal swabs from Gyeonggi and Gyeongbuk Provinces (Table 3).

A total of 2391 nasal swabs and lung tissues (355 farms) in pigs were tested for the detection of IDV. IDV RNA was not detected in samples obtained from pigs.

As can be seen in Figure 3, both antibody positive in cattle and pigs in Gyeongbuk and IDV RNA positive in cattle were confirmed.

### 3.3. Phylogenetic Analysis

Sequencing of the HEF was attempted for all positive samples. The partial or full-length HEF gene segment sequences were obtained from one cow nasal swab each from seven independent submissions/cases. The HEF segment was sequenced from the samples clustered within the D/Yamagata/2019 lineage. Alignment analyses showed that sequences of the HEF full gene found in this study shared the highest nucleotide similarity to that of the representative strain (D/bovine/China/JY3001/2021, accession number: ON415263-ON415269) of D/Yamagata lineage (99.35% nucleotide identity).

Phylogenetic analysis was performed using the maximum likelihood method in MEGA 11 for the HEF full gene. The analysis result also suggested that IDV in the ROK belongs to the D/Yamagata/2019 lineage (Figure 4).

## 4. Discussion

IDV or anti-IDV antibodies have been detected globally in various animal species including cattle, pigs, feral pigs, sheep, goats, horses and camelids. Human seropositivity has also been reported [28,29]. Since 2011, IDV has been identified in cattle species from parts of the United States as well as other countries such as Mexico, China, Japan, the United Kingdom, France, Italy, Ireland and Canada [20,30,31,32]. This first IDV surveillance study in the ROK found that in 2022, the seroprevalence of IDV in cattle was 54.6% (herd prevalence 56.5%) and higher than 1.4% (herd prevalence 14.8%) in pigs.

The findings in the ROK are similar to those of other countries. National serosurveillance conducted in the United States showed that for cattle, seroprevalence from 2014 to 2015 was 77.5%, and the seropositivity rate of pigs in 2011 was 9.5%, lower than compared to cattle [7,33]. Although the IDV viral RNA was detected in 4.8~18.0% of cattle in the United States from 2013 to 2014, it was only detected in 0.07% of cattle in 2016 [17,34,35]. In Europe, IDV seropositivity was 31.0~94.6% in cattle and 0~11.7% in pigs in 2019 [25,36,37]. From 2011 to 2016, IDV viral RNA positivity rates were 4.5~9.5% in cattle and 0.7~2.5% in pigs in Europe. From 2010 to 2016, the total positivity rate in cattle serological surveys in Japan was 30.5% [17,20,25]. In 2016, China’s IDV viral RNA positive rate in cattle was 7.9%, and 18.4% in pigs [16,17]. In 2016, the positive rate in Japan was 2.1% [17,38]. In most countries, epidemiological studies revealed that cattle had a higher prevalence compared to other species [36,39,40].

In the United States, 57 (19.1%) of 256 feral pigs tested positive for IDV, and viral RNA were found in respiratory epithelial cells, including the lungs [41]. Viral RNA and antibody levels in feral pigs were all negative in the ROK. However, the number of samples is likely to be too small to conclude with confidence its absence, and continuous monitoring will be required for feral pigs.

In 2022, IDV RNA was detected in 1.4% of cattle and 0% of pigs in the ROK, according to our study. In comparison to other countries, the low antibody titers of HI in cattle and pigs, as well as the 0% viral RNA detection rate in pigs, may indicate that IDV introduction and spread in the ROK may be comparatively recent when compared to other countries with previously reported IDV infection. IDV is not yet widespread in the ROK in pigs. In Jeju Island, no antibody or viral RNA to IDV were found in either cattle or pigs. Jeju Island is an autonomous administrative city, and the inflow of pigs and cattle is controlled and regulated, with limited number of livestock entering Jeju Island from the mainland. For this reason, IDV may not have been introduced into Jeju Island, hence the negative detection in our study. However, since the number of samples is smaller than those of other regions, further study will be needed.

Based on this study, more detailed and structured study is to be to be conducted in 2023 with the aim of elucidating epidemiological characteristics and possible patterns of risk, such as by region, livestock density, type of holdings, etc. This will include increasing the number of samples collected compared to the 2022 study and using a structured sampling scheme to allow for statistical analysis.

Six IDV genetic lineages have been identified based on the HEF gene. So far, the D/OK lineage is the most frequently reported lineage worldwide, and the D/660 lineage is mostly found in Europe, China and North America. In Japan, IDV lineages appear to be restricted to D/Yamagata/2016 and D/Yamagata/2019 [6]. Recently, D/CA2019, a novel phylogenetic lineage of IDV with broad antigenicity, was discovered in California, United States [42]. Despite the absence of an isolated virus in the Turkish cattle population, genomic characterization of IDV sequences revealed the existence of two genetic strains (D/Bursa2013 and D/Yama2019) [6]. IDV viruses circulating in Japan are genetically distinct from those in other geographical regions. D/Yama2019-lineage-like IDVs had been isolated only in Japan and Turkey until it was recently detected in nasal swab samples from healthy cattle in a Chinese abattoir in 2021 [42]. Our study also reports the detection of D/Yamagata/2019-like IDV in the ROK in 2022, thus expanding the known distribution of D/Yamagata/2019-lineage-like IDV to Japan, Turkey, China and the ROK. Studies will be needed to investigate the possible route of introduction into the country considering that the import of live cattle and pigs is prohibited from these countries to the ROK. Furthermore, additional research in Asia may be warranted to determine the distribution of D/Yamagata/2019-like IDVs outside these four Asian countries.

IDV infection causes respiratory disease in cattle and contributes to the cattle respiratory disease complex (BRDC) [6]. There have been several reports of IDV RNA detection in nasal swabs or lung tissue with respiratory disease [17,30,35]. However, viral RNA detection was also reported in nasal swabs from slaughtered healthy cattle [42]. Our nasal swab samples were taken from healthy cattle taken at abattoirs. In addition, IDV RNA positivity was confirmed in healthy cattle as was reported in China.

Interestingly, high IDV seroprevalence has been detected in human populations that have had contact with cattle, implying that the virus might pose a risk of animal-borne infection [6,28]. IDV can be transmitted and spread through direct contact with ferrets and guinea pigs, which are known surrogate models for human influenza [21,35]. Importantly, these viruses can proliferate efficiently in human airway epithelial cells (hAECs), which serve as an in vitro human respiratory epithelial model. Molecular surveillance of respiratory pathogens using bioaerosol sampling in hospital emergency rooms in the United States revealed the presence of IDV and the detection of the IDV genome in nasal lavage samples from Malaysian pig farm workers [43,44]. In addition, when aerosol samples were collected over 150 min in a congested area of Raleigh–Durham International Airport, United States, 4 out of 24 samples tested positive for the influenza D virus [45]. So far, very little is known about this virus. As a result, more research, such as studies of animal-borne infections in humans, is required to confirm the significance and evolution of this new virus.

This is the first report of IDV detection in the ROK. These findings highlight the importance of continuous surveillance in determining the distribution pattern of IDV in the ROK. Future studies are aimed at gaining an improved understanding of the epidemiology of IDV in the ROK.

## 5. Conclusions

Influenza A and B viruses have adversely affected animal and human health worldwide [2]. Previously, no research had been conducted in the ROK on the new influenza D virus, which is part of the *Orthomyxoviridae* family, and includes influenza A and B. Through our study, the distribution of influenza D virus in cattle and pigs in each province of the ROK was investigated for a 12-month period in 2022. The study demonstrated for the first time the presence of antibodies against IDV in cattle and pigs in the ROK. The antibody positive rate in cattle and pigs were 54.6% and 1.4%, respectively, showing stark difference between the species. IDV RNA was detected in seven samples from cattle, and sequence data acquired from samples indicated that the D/Yamagata/2019 lineage was circulating in the ROK. It is the same D/Yamagata/2019 lineage that has been reported in Japan and China, currently reported to be geographically restricted to Asia. The samples collected for this study will be helpful in conducting further investigations on IDV infection in the ROK. In addition, a more structured survey will be conducted in 2023 to better understand the epidemiological characteristics of IDV in the ROK, including possible risk factors and other determinants such as livestock density, farm type, etc. IDV was first detected in pigs in the United States, but subsequent studies have found that cattle are the predominantly affected livestock species. IDV is often reported to be a major cause of bovine respiratory disease, posing a significant health problem for the livestock industry worldwide. In particular, IDV is recognized as one of the causative agents responsible for the bovine respiratory disease complex, one of the most common and costly diseases in the United States. Furthermore, previous reports demonstrating that IDV can efficiently replicate in hAECs [43,44] suggests that IDV is a zoonotic disease with the potential to infect multiple species of animals. The ability of IDV in the ROK to cause disease in humans will also need to be further investigated in future studies.

## Figures and Tables

**Figure 1 microorganisms-11-01751-f001:**
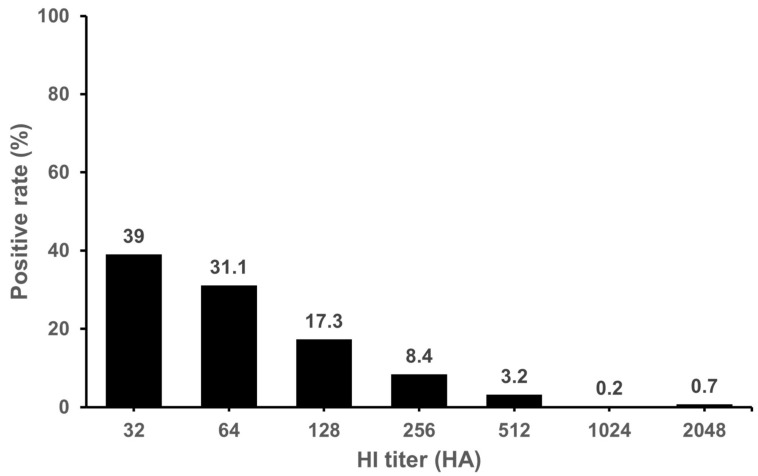
**Distribution of IDV positive antibody in domestic cattle.** The HI titer ≥ 20 was used as the cut-off value for seropositive samples. The highest positive rate HI titer was 32. Positive rate was confirmed at 39% in 32 HA and 31.1% in 64 HA. Cattle serum samples were not receptor-destroying-enzyme (RDE) treated.

**Figure 2 microorganisms-11-01751-f002:**
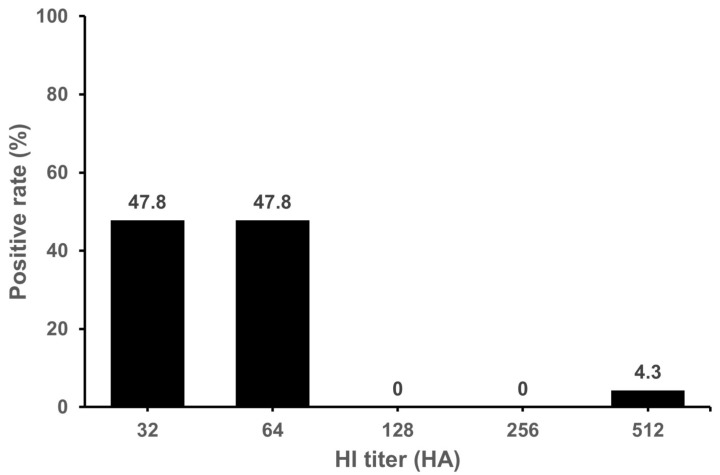
**Distribution of IDV positive antibody in domestic pigs.** The HI titer ≥ 20 was used as the cut-off value for seropositive samples. The highest positive rate HI titers were 32 and 64. A high positive rate was confirmed at 47.8% in 32 HA and 64 HA. Pig serum samples were pretreated overnight at 37 °C with receptor-destroying enzyme (RDE) at a ratio of 1:3.

**Figure 3 microorganisms-11-01751-f003:**
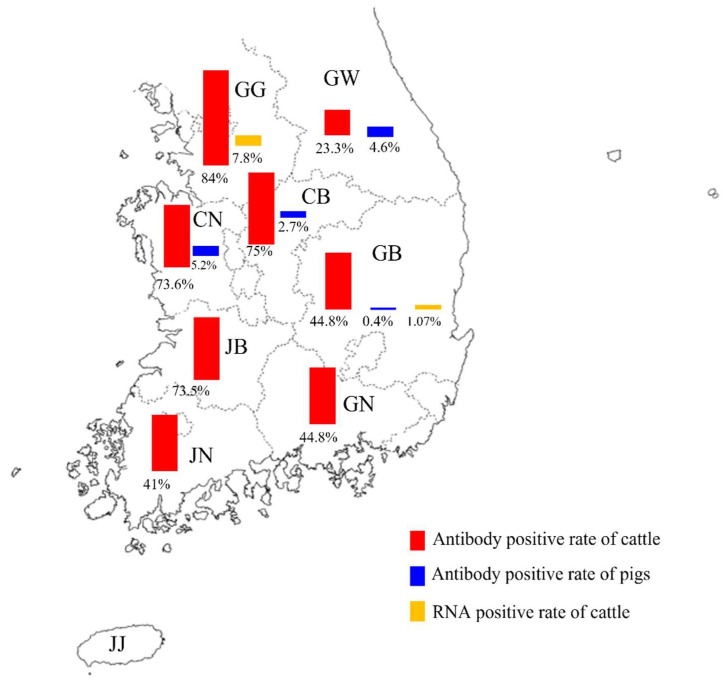
**Antibody and viral RNA positive rate of IDV in farmed cattle and pigs in each province of the ROK.** The red bars represent the antibody positive rate of cattle, blue bars represent the antibody positive rate of pigs and yellow bars represent the RNA positive rate of cattle. Gangwon, GW; Gyeonggi, GG; Chungbuk, CB; Chungnam, CN; Gyeongbuk, GB; Gyeongnam, GN; Jeonbuk, JB; Jeonnam, JN; and Jeju, JJ.

**Figure 4 microorganisms-11-01751-f004:**
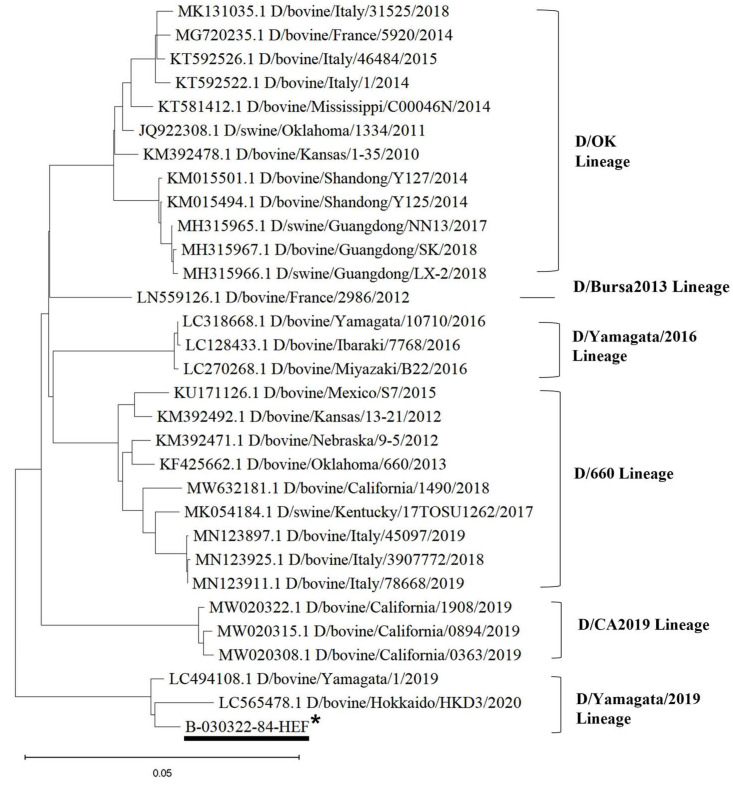
Phylogenetic tree of the HEF full gene of IDVs at the nucleotide level. * Sequence names in this study are underlined.

**Table 1 microorganisms-11-01751-t001:** Seropositivity rates and GMT of IDV in cattle in nine Provinces of the ROK.

Province *	Number of Tested Samples	Number of Positive Samples	Positive Rate (%)	GMT
Gyeonggi (GG)	119	100	84.0	70.0
Chungbuk (CB)	56	42	75.0	69.3
Chungnam (CG)	72	53	73.6	50.5
Jeonbuk (JB)	34	25	73.5	73.0
Gyeongbuk (GB)	261	117	44.8	55.0
Gyeongnam (GN)	58	26	44.8	82.1
Jeonnam (JN)	78	32	41.0	113.8
Gangwon (GW)	43	10	23.3	111.4
Jeju Island (JJ) **	21	0	0.0	0.0
Total	742	405	54.6	68.3

* There are nine provinces in ROK. ** Jeju island is a special autonomous island.

**Table 2 microorganisms-11-01751-t002:** Seropositivity rates and GMT of IDV in domestic pigs in nine Provinces of the ROK.

Province *	Number of Tested Samples	Number of Positive Samples	Positive Rate (%)	GMT
Chungbuk (CB)	80	5	6.3	84.4
Gangwon (GW)	80	4	5.0	45.2
Chungnam (CG)	259	12	4.6	45.3
Gyeongbuk (GB)	526	2	0.4	32.0
Gyeonggi (GG)	339	0	0	-
Gyeongnam (GN)	40	0	0	-
Jeonbuk (JB)	80	0	0	-
Jeonnam (JN)	41	0	0	-
Jeju Island (JJ) **	182	0	0	-
Total	1627	23	1.4	48.5

* There are nine provinces in ROK. ** Jeju island is a special autonomous island.

**Table 3 microorganisms-11-01751-t003:** Detection of IDV in nasal swabs and lung tissues in cattle and pigs.

Samples	Cattle		Pigs	
	Number of Tested Samples	Number of Positive Samples(Positive Rate %)	Number of Tested Samples	Number of Positive Samples
Lung tissues	45	0	159	0
Nasal swabs	954	14 (1.5%)	2232	0
Total	999	14 (1.4%)	2391	0

## Data Availability

The data presented in this study are available upon request from the corresponding author. The data are not publicly available due to patent application in the ROK.

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
