# Peer review of "First Detection of Influenza D Virus Infection in Cattle and Pigs in the Republic of Korea"

_microorganisms, 2023, doi:10.3390/microorganisms11071751_

Round 1
Reviewer 1 Report
Line 101: It will be good idea to inform the readers why bovine samples were not treated with RDE.
Line 134-135: Was there any statistical test performed?
Fig: Check figure numbers. First two figures have same number.
Table 3: It will be good to add some discussion points in line 267 - 271 about why lungs were negative and nasal swabs were positive.
Line 215: How was the herd prevalence rate calculated?
Line 218-223: Please add references for US and Europe data.
Some improvement needed. Please grammar check the manuscript.
Reviewer 2 Report
The article presents the first investigation of IDV prevalence in the Republic of Korea (ROK), revealing a seropositivity rate of 54.7% in cattle and 1.4% in pigs. The geometric mean antibody titers (GMT) were calculated as 68.3 in cattle and 48.5 in pigs. While the viral RNA positive rate was 1.4% in cattle, no viral RNA was detected in pigs. Furthermore, the authors conducted phylogenetic analysis on selected RNA samples, specifically targeting the hemagglutinin-esterase-fusion (HEF) gene. The results showed that all analyzed sequences were found to belong to the D/Yamagata/2019 lineage.
In my opinion the article is very interesting and the study represents the initial description of viral RNA and antibody surveillance of IDV in the Republic of Korea (ROK).
I have no comments on the current content of the article; however, I would recommend expanding the results section by including some statistical analysis, even basic, considering the authors have a substantial amount of data available. In my opinion, without these analyses, the article appears to be quite lacking in substance.
Minor editing of English language is required.
Author Response
We added “ Data were recorded into a standard Excel spreadsheet in Microsoft Excel 2016 (Microsoft corporation, Seattle, WA, Unite State). The geometric mean antibody titer (GMT) was calculated to compare the level of HI titer in positive sera only. GMT was calculated by averaging the logarithms of the HI titer and then converting the mean to a real number. Positive rates of animals by region and species are the number of positive animals divided by the total number of tested animals. The apparent herd prevalence was calculated as the total number of positive farms divided by the total number of tested farms.” sentences in manuscript Line 146-153.
Round 2
Reviewer 2 Report
Thank you for implementing my suggestions. The article in its current form is ready for publication.